# A comparative analysis of dementia strategies of seventeen European Countries in the context of Glasgow Declaration and WHO's Global Action Plan

Smruti Bulsari[1,2]*, Nureen Izyani Hashim[3], Kiran Pandya[4¤], Russell Kabir[3]

1 Institute of Public Health and Wellbeing, University of Essex, Colchester, Essex, United Kingdom,
2 NIHR Applied Research Collaboration (ARC), Cambridge, United Kingdom, 3 School of Allied Health, Faculty of Health, Medicine and Social Care, Anglia Ruskin University, Norwich, United Kingdom,
4 Shrimad Rajchandra Institute of Management and Computer Applications, Uka Tarsadia University, Bardoli, Gujarat, India

¤ Current address: Sarvajanik University, Surat, Gujarat, India
* smruti.bulsari@essex.ac.uk

## Abstract

Dementia prevalence across the globe is in alarming proportion and it is even expected to rise in the future. The World Health Organisation (WHO) had declared dementia as a health priority, way back in 2009 and had recommended then, that at least high-income countries develop a dementia action plan, and other countries develop a national dementia strategy (NDS). Later in 2014, European Countries came together to sign the Glasgow Declaration and agreed to develop their respective NDSs. Yet, a few countries still do not have their NDS. Moreover, some countries do not have their NDS in English. This study attempts to compare the dementia strategies of 17 European countries, which has a comprehensive NDS in English language. The study further examines how well these NDSs comply with the Glasgow Declaration and the WHO's Global Action Plan guidance. Cluster analysis is undertaken to classify NDSs of these countries in terms of similarity in the content. Word clouds are used to get an overall idea about the clusterwise contents of the NDSs, and then an algorithmic approach to content analysis is applied for identifying the clusterwise key focus areas of these dementia strategies. A comparative analysis of these NDSs in the perspective of dementia prevalence, demographic profile, per capita gross domestic product (GDP) and predominant healthcare financing model. The findings suggest that irrespective of the prevalence, country's demographic profile, GDP per capita or the predominant financing model, dementia strategies primarily focus on "care", implying the compliance of post-diagnostic support. Germany addresses almost all the fundamental rights of the Glasgow Declaration. Countries differ in their compliance with the key action areas of the WHO's GAP, though post-diagnostic support is addressed by the dementia strategies across the clusters,

**Data availability statement:** The data underlying the results presented in the study are available from the website of Alzheimer Europe: https://www.alzheimer-europe.org/policy/national-dementia-strategies.

**Funding:** This study is funded by National Institute of Health and Care Research (NIHR) Applied Research Collaboration (ARC) East of England and Alzheimer's Society. The funders had no role in study design, data collection and analysis, decision to publish, or preparation of the manuscript.

**Competing interests:** I have read the journal's policy, and the authors of this manuscript have the following competing interests: One of the co-authors (RK) is currently serving on the editorial board (academic editor) of PLOS One. This does not alter our adherence to PLOS ONE policies on sharing data and materials.

whereas 3 out of 7 KAAs are not addressed by dementia strategies in any of the clusters.

## Introduction

Dementia is a neurodegenerative disorder and a life-changing health condition that influences a person's memory, cognition, language, and behaviour. It is progressive in nature and one of the major causes of disability among people aged 60 and above. Dementia affects not only the individual but everyone in the family, and more specifically, the main caregiver. Globally, it is the seventh leading cause of death [1]. Dementia prevalence across the globe was 55 million in 2019, which is an increase from over 36 million in 2010, and is expected to rise to 135 million in 2050 [2,3]. The World Health Organisation (WHO) declared dementia as a world health priority way back in 2009. WHO had then recommended high-income countries to develop a national dementia action plan, and low and middle-income countries to develop national dementia strategies with a focus on improving primary healthcare and community services [4]. Five years later, European Countries came together to sign the Glasgow Declaration in 2014 and agreed to develop their respective National Dementia Strategies (NDSs). In 2017, WHO, launched a Global Action Plan (GAP) on public health response to dementia, with an objective to improve the lives of people living with dementia (PLwD) and their caregivers, as well as reduce the impact of dementia by focusing on preventative measures [5]. Therefore, the NDSs of the countries selected for this study, are compared against these fundamental rights of Glasgow Declaration and the KAAs of WHO's Global Action Plan, to examine the extent of their compliance.

Despite this, not all European countries have a national dementia strategy (NDS)/ Action Plan. National Dementia Strategies of 28 countries are published on the website of Alzheimer's Disease International [9]. At this juncture, it would be incisive to look into the fundamental rights laid down by the Glasgow Declaration and the KAAs of the WHO's Global Action Plan.

### Fundamental rights of Glasgow Declaration and key action areas of the GAP

Glasgow Declaration was launched at the 24th Annual Conference of the Alzheimer Europe, 2014, and it was unanimously adopted by 36 member countries. Alzheimer Europe is a non-profit, non-governmental organisation (NGO) comprising 41 national Alzheimer Associations, comprising 44 countries of Europe. It works towards improving lives of people living with dementia and their caregivers, through change in policy and practice, as well as changing society's perceptions to combat the stigma against dementia. Glasgow declaration called upon devising an umbrella dementia strategy of the entire Europe and also individual national dementia strategies of the member countries [5].

The fundamental rights laid down in the Glasgow Declaration are, the rights to:

• timely diagnosis,

- access to post-diagnostic support,

- person-centric, and coordinated care,

- equitable access to treatment and therapeutic interventions, and

- respect as an individual in their community.

WHO launched the GAP in 2017, with an objective to provide a blueprint for the Governments to improve the quality of life of PLwD and their cares. The key action areas of the WHO's Global Action Plan are:

1. Dementia as a public health priority,

2. Developing awareness about dementia and taking dementia-friendly initiatives,

3. Reduce the risk of dementia incidence,

4. Timely diagnosis, support, treatment, and care,

5. Providing necessary support and training to the caregivers of people with dementia,

6. Develop a core set of dementia indicators and collect data on those indicators, every two years, and

7. Undertaking research and focusing on innovations to improve the lives of people living with dementia.

The GAP is not legally binding. Yet, the ministries of all member states are obliged to provide regular progress reports to the WHO [6]. Arthurton et. al. [7] observe that regular monitoring of the work has resulted in more countries coming up with dementia strategies.

## Justification for this study

Despite signing the Glasgow Declaration in 2014, and the GAP following in 2017, 16 out of 44 European countries still do not have an NDS. On the other hand, England, France, Cyprus, Denmark, Belgium, Israel, Norway and Scotland had dementia strategies even before 2014, and most of these countries revised their NDSs after the Glasgow Declaration. It may be noted that while Israel is geographically in Asia, it is an associated state of the European Union and is also a member of Alzheimer's Europe.

There is a dearth of scientific literature analysing the contents and coverage of these dementia strategies, let apart its focus on examining the extent to which, these NDSs are informed by the Glasgow Declaration or the GAP. Some literature on comparative analysis of dementia strategies could be found in the context of Canada and its provinces [7], or in general, countries across the globe [8], but largely, in the context of the future NDS of Canada.

Other comparative analyses literature focus on specific areas like palliative care [10], person-centred care (PCC) [11], overall care [12], interaction between public health systems and social care systems, coordination in transition from home to residential care settings, and dementia care professionals [13] or human rights [14]. Alzheimer Europe already has a system of doing comparative analysis of dementia strategies their member countries [15,16].

Therefore, this study attempts to contribute to the scientific literature by undertaking a comparative analysis of the contents of NDSs of European countries and examine the extent to which these comply with the fundamental rights laid down in the Glasgow Declaration, as well as the KAAs of the WHO's Global Action Plan.

## Objectives of this study

1. To make a comparative analysis of the contents of the national dementia strategies of selected 17 European countries.

2. To examine the extent to which the dementia strategies comply with the fundamental rights of the Glasgow Declaration and WHO's GAP.

## Methods: Selection of the countries, framework and approach, and fiscal contextualisation of NDSs

### Data for this study: Sources and selection of the countries

The data for this study is the corpus of dementia strategies of selected European countries. Dementia strategies are downloaded from Alzheimer Europe [6], using a web-scraping algorithm developed by SB in Python 3.11.

Alzheimer Europe website [8] lists 44 European countries. However, sixteen European countries (Bosnia and Herzegovina, Bulgaria, Croatia, Estonia, Hungary, Jersey, Latvia, Lithuania, Montenegro, North Macedonia, Poland, Romania, Serbia, Slovakia, Turkey and Ukraine) do not have a dementia strategy, whereas Armenia and Portugal do not have a comprehensive dementia strategy that can be included for comparative analysis. Additional 9 European countries (Belgium –Wallonia, Czech Republic, France, Iceland, Italy, Luxembourg, Spain, Sweden and Switzerland) have NDSs in languages other than English; Belgium has two separate dementia strategies for Flanders and Wallonia, and the one for Flanders is in English. United Kingdom has separate NDSs for England, Scotland, Wales and Northern Ireland. Thus, a total of 17 NDSs, which are comprehensive and in English language, are analysed. These 17 countries, whose NDSs are analysed for this study are: Austria, Belgium (Flanders), Cyprus, Denmark, England, Finland, Germany, Gibraltar, Greece, Ireland, Israel, Malta, Netherlands, Northern Ireland, Norway, Scotland and Wales.

England, Belgium (Flanders), Gibraltar, Norway and Scotland have revised their NDSs over time. For the countries that have revised their dementia strategies over time, only the latest dementia strategies are included for the analysis. This is because, the objective of this study is to make an assessment of the current NDSs complying with the fundamental rights of the Glasgow Declaration of the KAAs of the Global Action Plan, which makes analysis of the latest dementia strategies relevant. Therefore, this makes it difficult to explicitly specify the time-period of this study. Though, the oldest dementia strategy is of Cyprus, which is published in 2012 and the most recent is Germany's, published in 2023; the dementia strategies used in this study were downloaded in November 2023; the year for the NDS of each of these countries are given in are S1 Table.

It may be noted that England refers to it as Prime Minister's Challenge to Dementia, Finland as the Memory Programme, and Belgium, Cyprus, Greece, Norway, and Wales as Action Plans. All national-level dementia strategies/ action plans, by whatever name, are now on, referred to as "National Dementia Strategy(ies)" or "NDS".

### Framework and approach: Comparative policy analysis

Comparative policy analysis is concerned with examining and understanding variations in the outcomes of governmental activities over time and jurisdictions. It attempts to answer questions related to the influence of social, economic and political situations on the contents and provisions of the public policies [9]. This implies that if there are differences in socioeconomic and political situations across countries, it would be reflected in the differences in their respective public policies. Out of the 17 European countries analysed for this study, 14 are members of the European Union (EU). Since EU directives promulgate policies in their member countries, homogeneity is expected across these NDSs. These countries are also member states of Alzheimer's Europe. Therefore, the NDSs of these countries are expected to be guided by the fundamental rights laid down in the Glasgow Declaration of Alzheimer Europe [5].

The *comparative method* of policy analysis is employed in this study, which helps designing a framework for describing the content, timings and outcomes of policies of different countries [10]. Peters [11] suggests the use of this method for *prima facie* classification of similar policies in groups. The similarities and differences in the policy text are explored using natural language processing (NLP) tools. There is a growing acceptance and use of NLP tools for policy analysis [12–14].

The NDSs are classified based on similarities (and dissimilarities) in their contents by applying k-means cluster analysis method. Lie et al. [15] suggest the use of k-means cluster over other methods of classification for its efficiency.

Dementia strategies across these European countries are cleaned and sentence tokenised before applying the k-means cluster analysis algorithm. Cleaning refers to removal of header/ footer text, table of contents, tables containing numbers and other text that can bias the analysis. The process of separating each sentence of the text corpus is called sentence tokenising. These sentences are then fed to the algorithm, Elbow method, to identify the optimal number of clusters "k" in which these NDSs will be categorised. Elbow is one of the most commonly used methods of determining the optimal clusters and provides a visual representation [19,20].

Word frequencies are generated to examine the focus areas of dementia strategies in each of these clusters. Word frequencies are often represented as word clouds, which is one of the static methods of summarising the text [16]; larger font size represents higher frequencies. While single words would give a broad idea about the focus of dementia strategies, using a set of two (or three) consecutive words, called bigrams (or trigrams) in NLP terminology, would help contextualise the use of words [17]. Details of pre-processing the text for required contextualisation is given in supporting information (S1 Text). This process would be similar to undertaking content analysis, except that an algorithmic technique, coded in Python 3.11 is used. A list of the top 30 bigrams and 20 trigrams is prepared, in the descending order of their frequency. Bigrams with similar meanings are combined into one theme; trigrams are combined in the same manner. For example, for cluster 1 countries, the bigrams containing the terms "care", "support" and "service" are treated as synonymous for identifying the key themes. These terms are found together with "health", "social", "people", "home" and "patient". Therefore, this theme is labelled broadly as "care for people with dementia". The frequency of occurrence of bigrams and trigrams in the clusterwise text of dementia strategies, and clubbing of terms therein, can be found in the supporting information (S2 and S3 Tables).

The advantage of using NLP techniques is that a large quantum of text can be analysed without the possibility of missing or overlooking any part of the text. It eliminates biases that might enter while interpreting the text manually. Moreover, manual interpretation of texts usually ranges over days, sometimes even with some breaks, which might lead to a change in the context of interpreting the text, leading to comparability issues. There is no subjectivity involved when using the NLP tools, and the algorithm takes several minutes to several hours, depending upon the quantum of text and the processor specifications. The disadvantage on the other hand is, since these techniques are based on word frequencies, the words with very low frequencies would not appear, despite being there in the text.

### Fiscal contextualisation of NDSs

Fiscal contextualisation is helpful in understanding why policies differ across countries. In our study, it would be helpful to understand the reasons of deviations, or partial non-adherence to either the fundamental rights of Glasgow Declaration or GAP. Fiscal contextualisation is undertaken by examining different models of healthcare services across the European countries included in this study, and their GDP, at purchasing power parity (PPP). Additionally, the clusters are examined in the context of dementia prevalence, percentage of population above 65 years of age, and life expectancy at birth for understanding the fiscal contextualisation of NDSs.

### Results: Dementia strategies classification

The output of the Elbow method in Fig 1, suggests four clusters, with the elbow appearing at k = 4. Thus, k is specified to be equal to 4 to classify the NDSs of these 17 countries.

These four clusters, with the countries' cluster membership, are shown in Table 1, where, the countries within each cluster have similarities in the contents of their NDSs:

Table 1 shows that cluster 2 and 3 are single country clusters comprising Germany and Norway respectively. As discussed earlier, the patterns in population, percentage of population above 65 years of age, life expectancy at birth, per

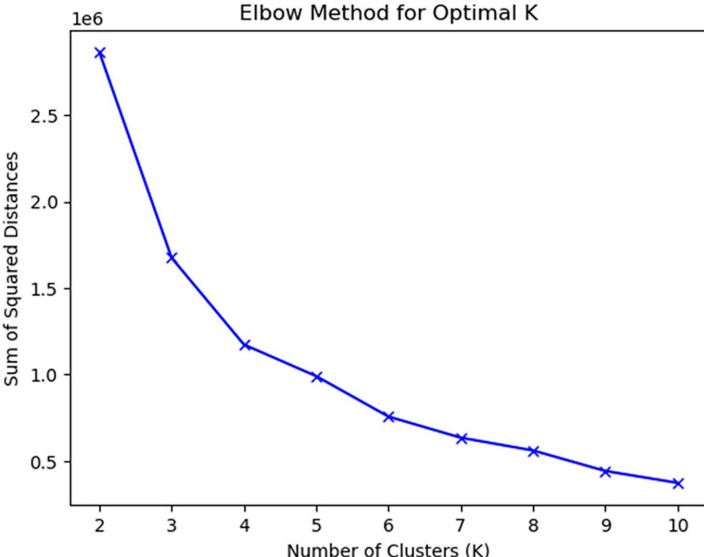

**Fig 1. Sum of Squared Distances for Different Values of "k": Elbow Method.**

**Table 1. Cluster-Membership of Countries based on their NDSs contents.**

| Cluster No. | Countries | Number of Countries in each cluster |
|---|---|---|
| 1 | Austria, Netherlands, Israel, Scotland, Gibraltar, Wales, Denmark, Cyprus, Belgium (Flanders) and Finland | 10 |
| 2 | Germany | 1 |
| 3 | Norway | 1 |
| 4 | England, Ireland, Malta, Northern Ireland and Greece | 5 |

capita income, and dementia prevalence across these clusters are examined for fiscal contextualisation. The summary measures of these indicators are shown in Table 2, and the details on these indicators for each of these 17 countries are given in S1 Table. It may be noted that the data on these indicators are available as consolidated figures for all the constituent countries of the United Kingdom. It was possible to find the data, separately for these countries, but there were comparability issues and hence, only consolidated figures are reported here. The statistics for the UK are excluded in estimating the summary measures, because Scotland and Wales are in cluster 1, whereas England and Northern Ireland are in cluster 4.

Table 2 shows that Germany, the only country in cluster 2, differs from the rest of the clusters in terms of having a very high dementia prevalence and having the highest population among all the 17 countries included in this study.

Norway, the only country in cluster 3 differs from the rest of the clusters in terms of highest per capita GDP, among the countries included in this study.

There is no apparent difference in the country profile indicators for clusters 1 and 4. The patterns in healthcare financing system in each of these countries is available in S1 Table.

Eight out of 17 countries have adopted the Beveridge model of healthcare financing, and five have adopted the Bismarck model; healthcare is financed by the government through tax payments in the Beveridge model, whereas it is financed by the insurance system, with joint contribution of the employers and the employees, in Bismarck model [18].

**Table 2. Clusters and Summary Measures of Dementia Prevalence, Demographic Profile and GDP Per Capita.**

| Summary Measures/ Actual Values | Dementia Prevalence | Population (in millions) | Percentage of Population above 65 years of age | Life Expectancy at Birth | GDP per capita PPP (const. 2017 USD) |
|---|---|---|---|---|---|
| **Cluster 1** | | | | | |
| Min | 0.94 | 0.03267 | 11.93 | 79.33 | 42379.16 |
| Max | 1.74 | 17.5017 | 22.89 | 82.26 | 58802.96 |
| Average | 1.43 | 7.45 | 18.26 | 81.42 | 50934.4 |
| Std. Dev. | 0.27 | 5.27 | 3.34 | 0.85 | 5949.94 |
| **Cluster 2:** Germany (Actual Values) | 1.91 | 83.40856 | 22.17 | 80.63 | 53395.65 |
| **Cluster 3:** Norway (Actual Values) | 1.41 | 5.403021 | 18.10 | 83.23 | 65915.53 |
| **Cluster 4** | | | | | |
| Min | 1.09 | 0.526748 | 14.83 | 80.11 | 29630.93 |
| Max | 1.99 | 10.44537 | 22.51 | 83.78 | 104671.9 |
| Average | 1.44 | 5.32 | 18.47 | 81.96 | 60300.49 |
| Std. Dev. | 0.38 | 4.06 | 3.14 | 1.5 | 32130.9 |

The Netherlands follows a mix of Bismarck and private voluntary insurance. Cyprus follows a hybrid of Bismarck and Beveridge models; it is a mix of government funding and mandatory social insurance contributions [19]. The Gibraltar Health Authority uses a healthcare model closely linked with that of the National Health System (NHS) in the United Kingdom, thereby following the Beveridge model [20], and Israel follows the Bismarck model of healthcare financing [21].

Taking this information into account, 3 out of 10 countries of cluster 1, follow the Bismarck model, 2 follow Bismarck + Beveridge (hybrid) and 5 follow the Beveridge model of healthcare financing. In cluster 4, 4 out of 5 countries follow the Beveridge model of financing; only Greece follows the Bismarck model. Thus, cluster 4 is largely homogeneous in terms of countries following the Beveridge model. This distinguishes cluster 4 from cluster 1.

## Major highlights of dementia strategies by clusters

The highlights of dementia strategies of each cluster are represented in the form of word clouds in Fig 2–5.

The word cloud of Fig 2 shows that "support", "care", "service", "need" and "people" are the top five frequently occurring terms in the combined text of the dementia strategies of cluster 1. Terms "support" and "service" can be interpreted to a part of the broader concept "care". These five terms together would mean the focus of these dementia strategies is on "care for people (with dementia) in need". Since "dementia" and "Alzheimer's Disease" are removed from the list of words to be counted, they do not appear in the word cloud. The context is nevertheless, "people with dementia" and hence it is mentioned in the brackets.

Fig 3 highlights the terms "end", "need", "support", "provide" and "people". The term "support" is interpreted as synonymous to "care", and the term "end" appears in the context of end-of-life. Therefore, the focus of the dementia strategy of Germany is on "providing end-of-life support to people (with dementia) in need".

Norway has a dementia action plan. Unlike strategies, action plans have targets set with timelines. Therefore, the term "plan" is one of the frequently occurring terms Fig 4. The top frequently occurring terms are "care", "service", "people", "need", "good" and "patient". This could be interpreted as "planning care for people and patients (with dementia) in need". The term "patient" is largely used to express illness or a person who needs consultation from a general practitioner or hospital services. On the other hand, dementia is a condition, though PLwD might require hospital care at advanced stages. People living with dementia might have to be hospitalised for reasons other than dementia-related issues too.

 

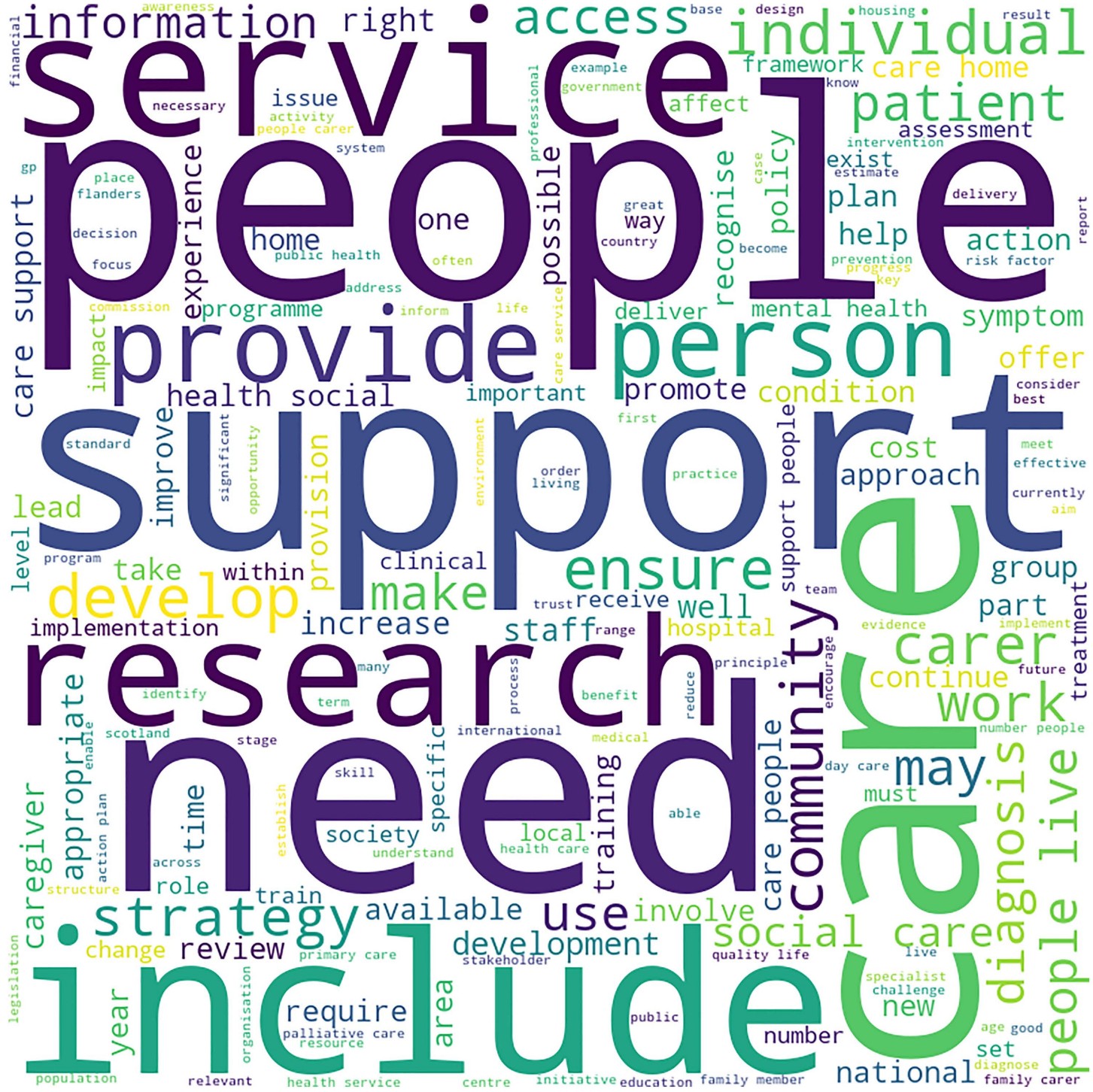

**Fig 2. Word Cloud for Dementia Strategy Text of Cluster 1 Countries.** Highlights these words in the combined text of dementia strategies of countries of cluster 1: "support", "care", "service", "need" and "people".

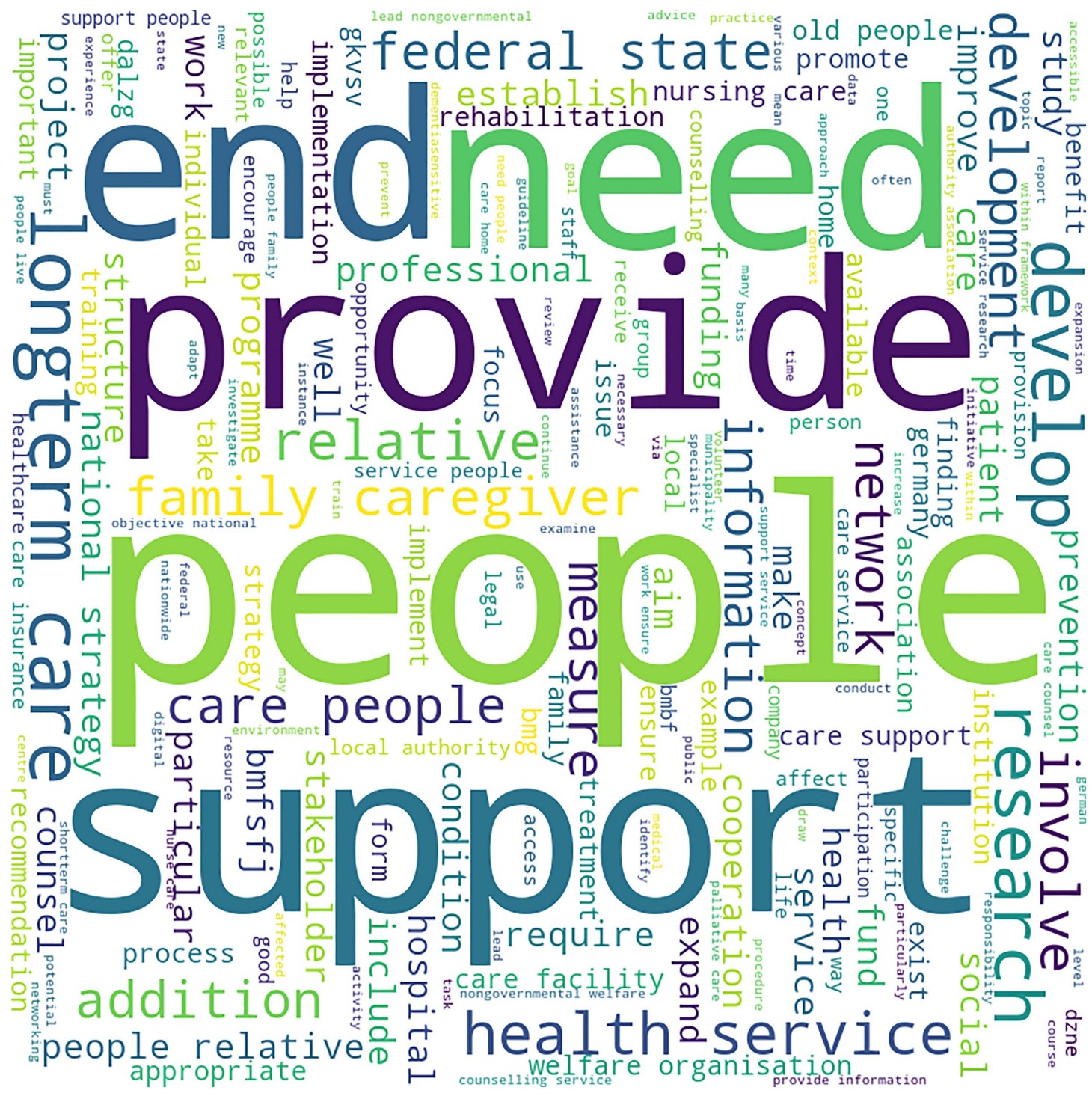

**Fig 3. Word Cloud for Dementia Strategy Text of Cluster 2 Country (Germany).** Highlights these words in the national dementia strategy of Germany, the only country in cluster 2: "end", "need", "support", "provide" and "people".

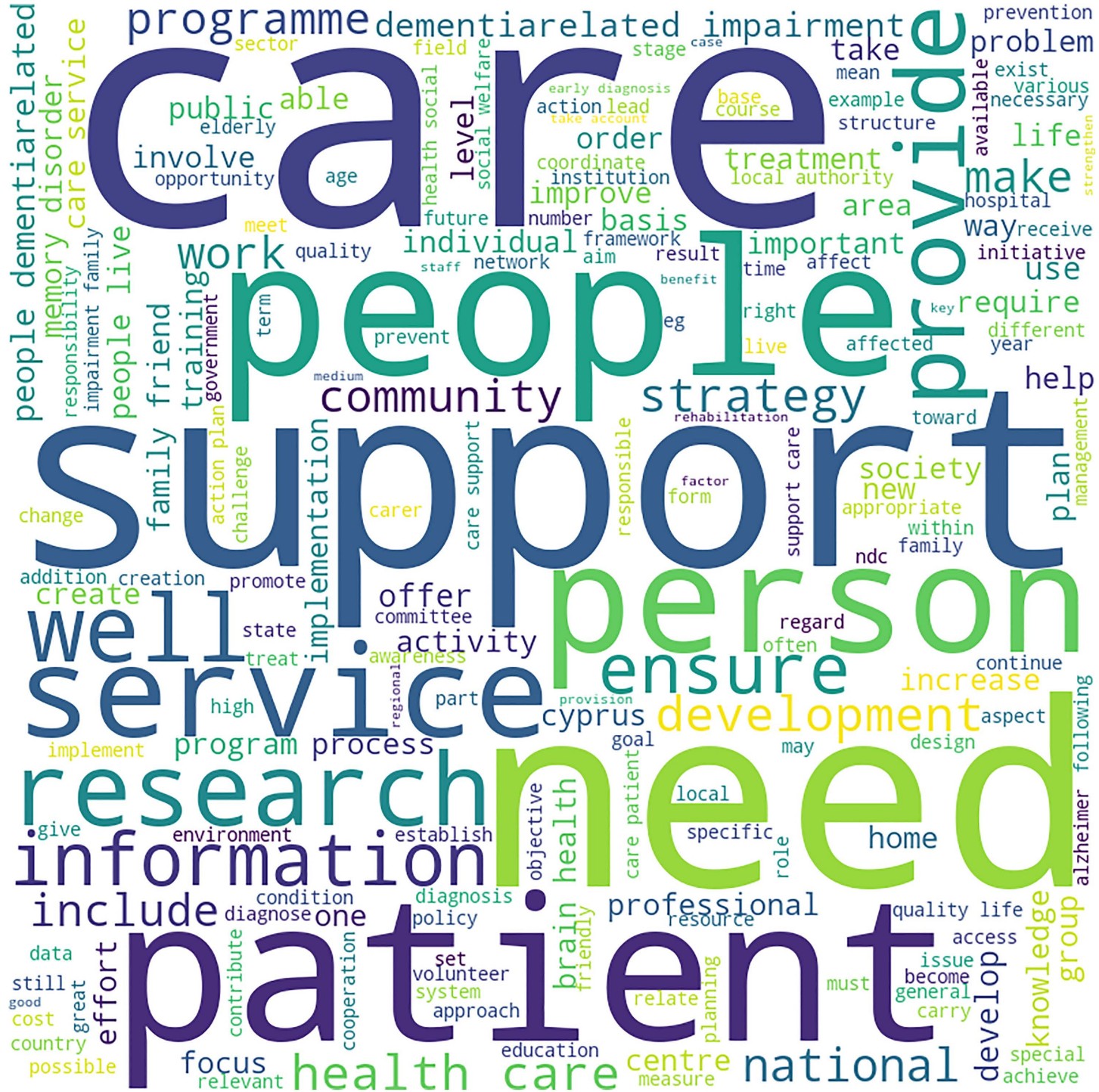

**Fig 4. Word Cloud for Dementia Strategy Text of Cluster 3 Country (Norway).** Highlights these words in the national dementia strategy of Norway, the only country in cluster 3: "care", "service", "people", "need", "good" and "patient".

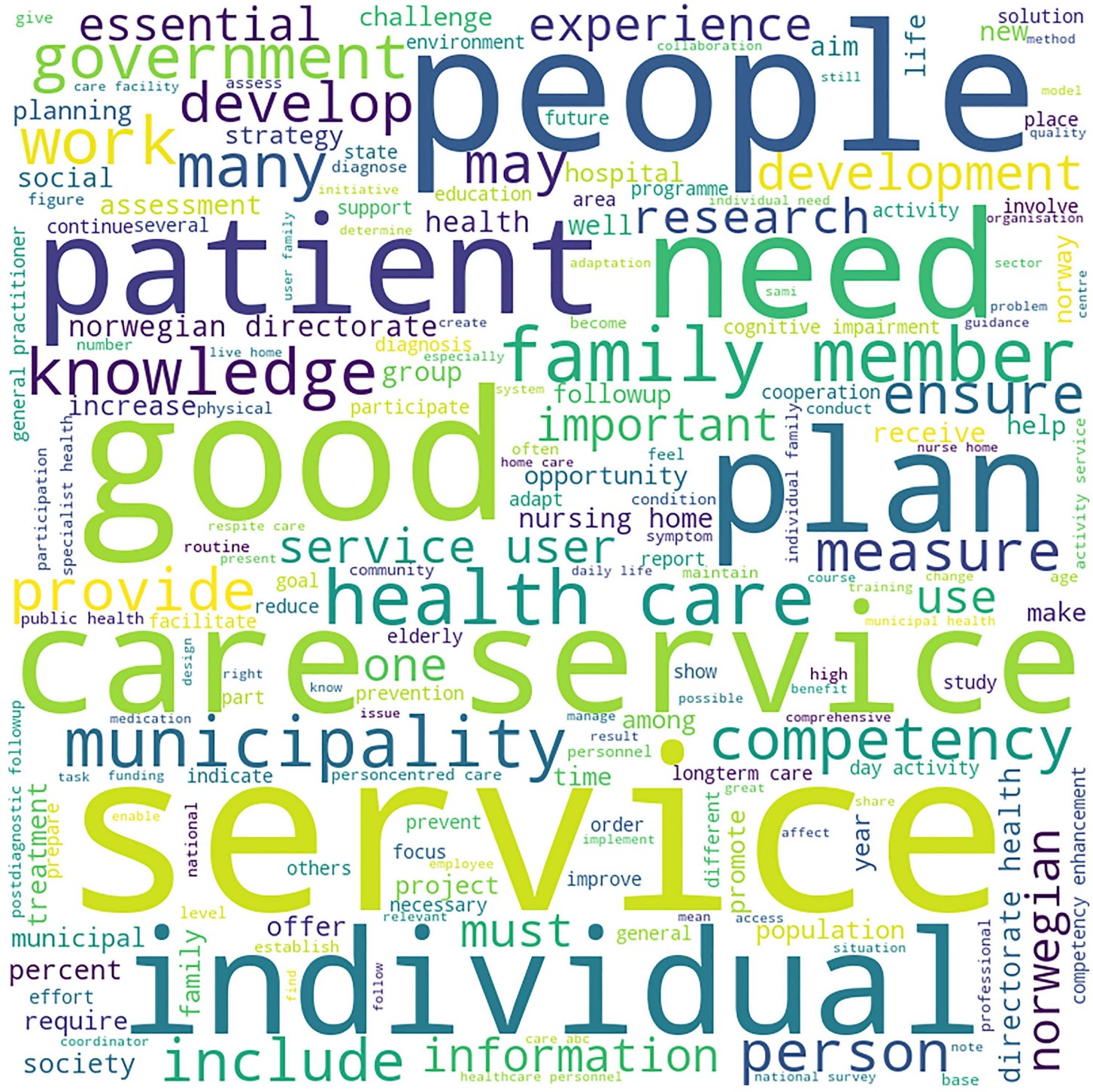

**Fig 5. Word Cloud for Dementia Strategy Text of Cluster 4 Countries.** Highlights these words in the combined text of dementia strategies of countries of cluster 4: "support", "service", "people", "include" and "need". The focus on "research" can also be seen in this word cloud.

Fig 5 also highlights similar terms as the other clusters: "support", "service", "people", "include" and "need". There is some emphasis on "research" too, in this word cloud. This can be interpreted as "care for people (with dementia) in need" and "(dementia) research".

All these word clouds show that "care for people with dementia" is the core focus area of all NDSs, irrespective of the cluster in which they belong to. However, additionally, the dementia strategy of Germany emphasises on end-of-life care and the Norway Action Plan provides for care for patients, who are living with dementia. Cluster 4 countries emphasise on research, over and above "care for people with dementia".

This implies that dementia strategies focus on post-diagnostic care, which is one of the fundamental rights of people living with dementia, as laid down in the Glasgow Declaration, and also one of the KAAs of the Global Action Plan.

This analysis is, however, based on individual terms. Contextualisation of these terms with the terms will give deeper insights into the key themes of dementia strategies.

**Key themes of dementia strategies across clusters**

Bigrams' frequencies reveal the following themes:

Cluster 1 (formed of 10 countries):

- Care for people with dementia, ranging from health care, social care, patient care, personal care and informal care. It may be noted that service is also interpreted as care in this context.

- People (with dementia), their family and friends.

- Quality of life of people with dementia and related impairment.

Cluster 2 (Germany):

- Care at different levels, for people with dementia: This includes long-term care, family care, nursing service, palliative care, short-term care, and care homes.

- Role of the State/ system: This includes the role of local authority, system of (health-related) insurance and other organisations for the welfare of people living with dementia.

- Services for people living with dementia, which include health services, counselling and providing information.

Cluster 3 (Norway):

- Care at different levels, for people living with dementia: This includes long-term care, person-centred care, respite care, and care facility.

- Role of family members in care for people living with dementia.

Cluster 4 (formed of four countries):

- Care at different levels, for people living with dementia: social care, primary care, health care, daycare, family care, individual care, professional care and care management.

- Services related to public health.

One may notice the frequency of trigrams is less as compared to that of bigrams. This is because as the contextualisation becomes more specific, the probability of repetition of chunks of words reduces. However, bigger chunks of words (trigrams) provide more specific contextualisation, and therefore, it is important to analyse trigrams as well. Trigrams' frequencies reveal the following themes:

Cluster 1:

- Care for people with dementia, especially social care and unpaid care, remains at the top of the themes of these dementia strategies.

- This is followed by dementia-related impairment, and its effect on family and friends.

    Cluster 2 (Germany):

- There is a strong emphasis on long-term care for people with dementia.

- This is followed by services and research related to health, hospice, and palliative care facilities. There is also a focus on the need for defining the term "care".

    Cluster 3 (Norway):

- Care for people with dementia at different levels: This includes health care, specialist services, municipal health care, long-term care facility, home care, respite care, daycare, and services to patients.

- The focus is on the State's system of healthcare. The Norwegian Dementia Action Plan focuses on having a national guideline, setting up a competency enhancement plan, and setting up a national advisory unit to facilitate services for people living with dementia.

    Cluster 4:

    This cluster has only one theme, which is care for people with dementia. This includes social care, family care, professional care, individual care, end-of-life care, daycare, and hospital care.

### Comparison of NDSs clusters with the fundamental rights of the Glasgow Declaration and the KAAs of the WHO's Global Action Plan

All the national dementia strategies fully address the "access to post-diagnostic support" right of the Glasgow declaration, whereas none of the NDSs address the "timely diagnosis" right. Person-centric care and co-ordinated care rights are addressed by clusters 1, 2 (Germany) and 3 (Norway), whereas the fundamental rights to "equitable access to treatment" and "respect as individual in their community" are addressed only by cluster 2 (Germany).

The KAA of "dementia as a public health priority" is addressed by all the NDSs, whereas the KAAs to "reduce the risk of dementia incidence", "providing necessary support and training to the caregivers of people with dementia", and "developing a core set of dementia indicators and collect data on those indicators, every two years" are not addressed by any of the NDSs. The "support" and "care" dimensions of "timely diagnosis, support, treatment and care" are addressed by clusters 1, 2 (Germany) and 3 (Norway). "Developing awareness about dementia" is addressed only by Germany and "undertaking research and focusing on innovations to improve the lives of people living with dementia" is addressed only by Norway.

A comprehensive comparison of different clusters' adherence to the fundamental rights of the Glasgow Declaration as well as the KAAs of the GAP are summarised in Table 3:

## Discussion

The analysis of the text of NDSs selected for this study shows that the major emphasis is on "care", which is confirmed by other studies too.

Though, Hampel et al. [22] and Alzheimer Europe [23] observe that the dementia strategies focus on risk reduction/prevention and early diagnosis. The results of the present study show that the NDSs the emphasis on risk reduction/dementia prevention and early diagnosis is a lot lower as compared to what is given to "care".

Existing literature suggest the need for the NDSs to focus on increasing awareness and reducing the stigma surrounding dementia, as well as promoting research on dementia [24–26]. *Albeit*, these studies are undertaken in the context of

**Table 3. An Overview of the Clusters' Adherence to the Fundamental Rights of Glasgow Declaration and the Key Action Areas of the WHO's Global Action Plan.**

| Fundamental Rights of the Glasgow Declaration | Cluster 1 | Cluster 2 | Cluster 3 | Cluster 4 |
|---|---|---|---|---|
| Timely diagnosis | | | | |
| Access to post-diagnostic support | ✓ | ✓ | ✓ | ✓ |
| Person-centric, and coordinated care | ✓ | ✓ | ✓ | |
| Equitable access to treatment and therapeutic interventions | | ✓ | | |
| Respect as an individual in their community | | ✓ | | |
| **Key Action Areas of WHO** | | | | |
| Dementia as a public health priority | ✓ | ✓ | ✓ | ✓ |
| Developing awareness about dementia and taking dementia-friendly initiatives | | P | | |
| Reduce the risk of dementia incidence | | | | |
| Timely diagnosis, support, treatment, and care | P | P | P | |
| Providing necessary support and training to the caregivers of people with dementia | | | | |
| Develop a core set of dementia indicators and collect data on those indicators, every two years | | | | |
| Undertaking research and focusing on innovations to improve the lives of people living with dementia. | | | ✓ | |

P = Partial adherence, ✓ = Full adherence.

A detailed discussion on part adherence of the KAAs can be found in the discussion section.

countries around the world, and they do not specifically study European countries. Though, the results of this study also support this finding.

Alzheimer Europe [23] observes that the most NDSs do focus on diagnosis and treatment, training of practitioners, raising awareness about dementia and undertaking research on dementia. However, the results of this study show that a lot more attention is required to be given to these provisions in the dementia strategies.

Existing literature on analysis of dementia strategies from across the World [26] show that the NDSs had guidance on involving people living with dementia (PLwD) and their caregivers' involvement in future plans, the strategies differed in their approach to involving PLwD in advanced stages and care for minority and disadvantaged groups, and people with early onset dementia.

On the other hand, the findings of Alzheimer Europe [27] show that while countries have been proactive in publishing the policies, little efforts have been made towards allocating finance for implementing these policies.

A discussion on the areas addressed and those requiring attention, in the context of the fundamental rights of the Glasgow Declaration and the KAAs of the WHO's Global Action Plan, are presented in the next two sections.

## NDSs clusters in the context of the fundamental rights of the Glasgow Declaration

Cluster 1:

The NDSs of the countries in this cluster focus on "care for people living with dementia", especially complying with the person-centred care and co-ordinated care rights of the Glasgow Declaration. Quality of life of PLwD, dementia-related impairment and its impact on carers highlight some concerns around the access to post-diagnostic support rights. However, there is to emphasise on the rights to timely diagnosis, equitable treatment and therapeutic interventions, and their respect as an individual in their community. Additionally, these dementia strategies express concerns and provide for the carers of PLwD.

Cluster 2 (Germany):

The national dementia strategy of Germany addresses the fundamental right to access to post-diagnostic support through its focus on the role of State to ensure the well-being of PLwD. The role of State to ensure the well-being of PLwD

also addresses the right to equitable access to treatment and therapeutic interventions. The right to access to post-diagnostic support is also evident through its focus on short and long-term care, family care as well as care homes, hospice and palliative care. This implies that the fundamental right to person-centric and co-ordinated care is also addressed. The NDS of Germany takes into consideration all types of care and support at all levels for PLwD. The services for PLwD addresses their fundamental rights to be respected as an individual in the community, by giving them a life with dignity. However, Germany too, needs to focus on the right to timely diagnosis.

Cluster 3 (Norway):

Care at different levels – long-term as well as short-term care, home care, respite care, and daycare for PLwD, augmented with the State's intervention in the form of having a national guideline for facilitating services to people living with dementia addresses the fundamental right to access to post-diagnostic support. There is a specific focus on person-centred care in the NDS of Norway, thereby addressing the "PCC" part of the fundamental rights, though the "coordinated care" part requires attention. The fundamental rights to timely diagnosis, equitable access to treatment and therapeutic interventions, and respect as an individual in the community are yet to be addressed.

Cluster 4:

The NDSs of these countries focus on care at different stages – ranging from social care, primary care, health care, daycare, family care, individual care, professional care, end-of-life care, hospital care and care management. This implies that these NDSs address the fundamental rights to post-diagnostic support. It may be noted that individual care is different from person-centred care (PCC). PCC is a holistic approach encouraging the person cared for, in making an informed decision about their possible care alternatives and beyond [28]. Thus, the NDSs of these countries need to focus on rest of the other fundamental rights – timely diagnosis, equitable access to treatment and therapeutic interventions, and respect as an individual in the community, over and above person-centred care and co-ordinated care.

## NDSs clusters in the context of the key action areas of WHO's Global Action Plan

The very existence of a national dementia strategy addresses the first KAA of the GAP: dementia to be accorded public health priority. As discussed in earlier, three KAAs out of seven - KAAs to "reduce the risk of dementia incidence", "providing necessary support and training to the caregivers of people with dementia" and "developing a core set of dementia indicators and collect data on those indicators, every two years" – are required to be addressed by all the NDSs. Out of the remaining three KAAs:

Cluster 1:

From the KAA on timely diagnosis, support, treatment and care, only "care" has been the focus of the NDSs of countries in this cluster. The KAA on timely diagnosis, support and treatment and developing awareness about dementia, taking dementia-friendly initiatives and undertaking research and focusing on innovations to improve the lives of people living with dementia require more attention.

Cluster 2 (Germany):

In the context of the KAAs of WHO's global action plan, focusing on care, at different levels, for PLwD addresses the "care" and "support" part of the timely diagnosis, support, treatment and care KAA. More attention is required on timely diagnosis and treatment aspect of this KAA. The role of State in care and providing support for PLwD, plus, providing for services and information, including counselling also addresses the "support" part of the timely diagnosis, support, treatment and care, as well as the "taking dementia-friendly initiatives" of the second KAA. The "developing awareness about dementia" part of the second KAA and "undertaking research and focusing on innovations to improve the lives of people living with dementia" requires to be addressed.

Cluster 3 (Norway):

Just like Germany's NDS, the NDS of Norway also emphasises the "care" and "support" parts of the timely diagnosis, support, treatment and care, by providing for different levels of care for PLwD at different stages of dementia, and an

active role of the State in facilitating services for PLwD. Norway's NDS however, focuses to quite an extent on undertaking research and focusing on innovations to improve the lives of people living with dementia. Timely diagnosis and treatment need to be emphasised in the forthcoming action plans for dementia. The KAA on developing awareness about dementia and taking dementia friendly initiatives requires attention.

Cluster 4:

The NDSs of countries in this cluster focuses largely on "care" and "support" parts of the timely diagnosis, support, treatment and care KAA of the Global Action Plan. The timely diagnosis and treatment parts, as well as the KAAs – on developing awareness about dementia and taking dementia friendly initiatives and undertaking research and focusing on innovations to improve the lives of people living with dementia require attention.

Irrespective of the clusters, there is a need for active participation of people living with dementia at all stages, in future dementia action plans of all the countries. Vinay & Biller-Andorno [26] observed that while the NDSs had guidance on involving people living with dementia (PLwD) and their caregivers' involvement in future plans, the strategies differed in their approach to involving PLwD in advanced stages and care for minority and disadvantaged groups, and people with early onset dementia.

## Financial implications of implementing the fundamental rights of Glasgow Declaration and the KAAs of the WHO's Global Action Plan

Providing for the Glasgow Declaration's fundamental rights or the KAAs of WHO's Global Action Plan have financial consequences. The financial burden of healthcare for the countries adopting Beveridge model falls on the State (via taxpayers), whereas for the countries adopting Bismarck model, it falls on individuals (employers and employees) through insurance contributions. This requires an understanding of different measurable and non-measurable costs of implementing each of these fundamental rights/ KAAs. There are significant regional variations in the types of costs, which makes it difficult for all the countries to address all the fundamental rights/ KAAs in one go. Priorities are required be accorded to address the fundamental rights/ KAAs by examining the cost-benefit/ cost-effectiveness analysis. There is dearth of literature on the cost-benefit/ cost-effectiveness analysis in the context of implementing the fundamental rights of Glasgow Declaration/ KAAs of the Global Action Plan. Therefore, a brief discussion on reducing the risk of dementia incidence and PCC is presented in this section.

**Reduce the risk of dementia incidence:** Multidomain interventions like diabetes prevention, behavioural change, food supplements programme, reducing cardiovascular risk, to name a few, are found to reduce dementia risk and improve the quality-adjusted life years (QALY) [29,30] as well as personal savings [31]; The Lancet Commission has identified 14 modifiable risk factors to prevent dementia. This makes a strong case for dementia strategies to focus on prevention for the countries adopting Bismarck model.

**Person-centred care:** There is evidence of improvement in incremental QALY, behavioural and neuropsychiatric symptoms as well as enhanced care quality through person-centred care [32,33]. PCC, by inclusion of psychodynamic perspective, is also found to enhance the cost-effectiveness of healthcare provision under resource constraints [34]and in organisational settings [35]. This area could therefore be prioritised by the countries opting Beveridge model.

Emphasis should also be on PCC in home care settings, irrespective of the healthcare financing model, though this is more suitable to the countries having a mix of Bismarck and Beveridge models.

## Limitations of this study

It may be noted that in this study, the key focus areas for each cluster are identified based on frequency of occurrence of a term or a combination of terms in the NDSs clusters. This in no way implies that there is no provision for other areas like prevention, raising awareness etc. It only means that other areas require higher attention that what is already accorded in the NDSs.

## Conclusion

Post-diagnostic support is addressed by dementia strategies of all the countries, with the only exception of Cluster 2 – Germany, which addresses almost all the fundamental rights of the Glasgow Declaration. However, different countries differ in their compliance with the key action areas of the WHO's GAP, though post-diagnostic support is addressed by the dementia strategies across the clusters. On the other hand, research and innovation is not addressed by dementia strategies in most of the clusters. Germany is found to focus on undertaking dementia-friendly initiatives. All the dementia strategies should consider involving PLwD in advanced stages to make the dementia strategies more inclusive.

Dementia prevention, person-centred care, increasing awareness, and reducing stigma around dementia, plus, devising a set of dementia indicators and regular collection of data on those indicators require attention in the national dementia strategies of most of these 17 European countries. Emphasising dementia prevention strategies could be prioritised by countries adopting Bismarck model, person-centred care could be prioritised by those adopting Beveridge model, whereas PCC in home care setting can be accorded priority by those adopting a mix of Bismarck and Beveridge models.

## Supporting information

**S1 Text.  Methodological note.**
(DOCX)

**S1 Table.  Countries by cluster membership, dementia prevalence, demographic profile, per capita GDP and healthcare financing model.**
(DOCX)

**S2 Table.  Bigram tables.**
(DOCX)

**S3 Table.  Trigram tables.**
(DOCX)

## Author contributions

**Conceptualization:** Smruti Bulsari.

**Data curation:** Nureen Izyani Hashim.

**Formal analysis:** Smruti Bulsari.

**Funding acquisition:** Smruti Bulsari.

**Methodology:** Smruti Bulsari.

**Project administration:** Smruti Bulsari.

**Resources:** Russell Kabir.

**Software:** Smruti Bulsari.

**Supervision:** Kiran Pandya.

**Writing – original draft:** Smruti Bulsari.

**Writing – review & editing:** Kiran Pandya.

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
