## [Decision Letter · Decision Letter 0]

25 Mar 2025

PONE-D-25-06352A Comparative Analysis of Dementia Strategies of Fifteen European Countries in the Context of Glasgow Declaration and WHO’s Global Action PlanPLOS ONE

Dear Dr. Bulsari,

Thank you for submitting your manuscript to PLOS ONE. After careful consideration, we feel that it has merit but does not fully meet PLOS ONE’s publication criteria as it currently stands. Therefore, we invite you to submit a revised version of the manuscript that addresses the points raised during the review process.

**ACADEMIC EDITOR: **

The authors have investigated an important area related to dementia care provision. Although comparative analysis of NDSs were evident in the manuscript, the academic manuscript writing style should have been improved. There should be a clear flow of information in the manuscript, starting from the introduction, justification for the current project, objectives, methodology etc. Although the authors mentioned the subtopics accordingly, the information flow is lacking. Information presented in a mixed manner which affects the readability and understanding.

Most of the article description have used first-person pronoun ‘We’. Using first-person language may detract from the arguments and/or viewpoints present in the writing. Therefore, avoid using first-person frequently, you may use passive voice instead

Eg. ‘*We apply k-means cluster analysis to classify the NDSs based on similarities’*

Suggest rewording as follows.

NDSs based on similarities were classified using k-means cluster analysis.

The above changes and the changes suggested by reviewer 1 should be addressed in order to consider for publication in PLOS One.

We look forward to receiving your revised manuscript.

Kind regards,

Surangi Jayakody, MBBS, MSc, MD

Academic Editor

PLOS ONE

Journal Requirements:

I have read the journal's policy and the authors of this manuscript have the following competing interests: One of the co-authors (RK) is currently serving on the editorial board (academic editor) of PLOS One.

This study is funded by National Institute of Health and Care Research (NIHR) Applied Research Collaboration (ARC) East of England and Alzheimer's Society

Smruti Bulsari, University of Essex, is supported by the National Institute for Health and Care Research – Applied Research Collaboration East of England (NIHR ARC EoE) and the Alzheimer’s Society, funded through a Post-Doctoral Fellowship. The NIHR ARC EoE is hosted by Cambridgeshire and Peterborough NHS Foundation Trust. The views expressed are those of the authors and not necessarily those of the NIHR or the Department of Health and Social Care.

This study is funded by National Institute of Health and Care Research (NIHR) Applied Research Collaboration (ARC) East of England and Alzheimer's Society

Reviewers' comments:

Reviewer's Responses to Questions

**Comments to the Author**

1. Is the manuscript technically sound, and do the data support the conclusions?

Reviewer #1: Partly

Reviewer #2: Yes

2. Has the statistical analysis been performed appropriately and rigorously? 

Reviewer #1: N/A

Reviewer #2: Yes

3. Have the authors made all data underlying the findings in their manuscript fully available?

Reviewer #1: No

Reviewer #2: Yes

4. Is the manuscript presented in an intelligible fashion and written in standard English?

Reviewer #1: No

Reviewer #2: Yes

5. Review Comments to the Author

Reviewer #1: Dear editor,

Thank you for the opportunity to review the manuscript titled "A Comparative Analysis of Dementia Strategies of Fifteen European Countries in the Context of Glasgow Declaration and WHO’s Global Action Plan".

The authors have attempted to investigate the important issue of how well the national dementia strategies (NDSs) of 15 European countries align with the Glasgow Declaration and the WHO’s Global Action Plan guidance. However, the current version of the article does not meet academic standards, as information relevant to the introduction, methods, and results is interwoven across all sections. For example, the introduction includes methodological details, and vice versa. I strongly recommend that the authors carefully revise the text to ensure that each section is structured appropriately.

Although the primary objective of the study is to examine how well the selected NDSs comply with the Glasgow Declaration and the WHO’s Global Action Plan, the article does not directly compare the NDSs with these benchmark criteria. Furthermore, the discussion is not structured around the key action areas outlined in the WHO framework or the Glasgow Declaration.

It is also unclear why more than half of the discussion focuses on the costs and cost-effectiveness of dementia interventions, given that these were not the primary focus of the study. Additionally, a clear summary of findings is missing, and the results are not interpreted in relation to the WHO key action areas or the Glasgow Declaration.

Therefore, I strongly recommend authors to revise the manuscript to address these major issues.

Reviewer #2: Authors presented comparative analysis of the Dementia Strategies of 15 European countries in the context of Glasgow Declaration and Global Action Plan of World Health Organization. In the article, comparative analysis of the National Strategic Plans of Dementia in the fields of prevalence, demographic profile, GDP and financial model have been done and presented and based on the findings of the study, recommendations suggested.

In this article the authors used a good scientific writing style which is attractive, and the flow of the information is good. The title is appropriate. The structure is clear and presented concisely.

However, the tables should be formatted.

In table 2, life expectancy at birth (in years) could be added.

Though the discussion is concise, it addresses the areas of interest.

6. PLOS authors have the option to publish the peer review history of their article (what does this mean? ). If published, this will include your full peer review and any attached files.

**Do you want your identity to be public for this peer review?** For information about this choice, including consent withdrawal, please see our Privacy Policy .

Reviewer #1: No

Reviewer #2: No

---

## [Author Response · Author response to Decision Letter 1]

4 May 2025

Editorial Comments:

Comment 1:

There should be a clear flow of information in the manuscript, starting from the introduction, justification for the current project, objectives, methodology etc. Although the authors mentioned the subtopics accordingly, the information flow is lacking. Information presented in a mixed manner which affects the readability and understanding.

Response 1:

The manuscript is restructured to ensure the clear flow of information, and it is ensured that there are no overlaps in introduction, justification for the current project, objectives, methodology, results and discussion sections.

Comment 2:

Most of the article description have used first-person pronoun ‘We’. Using first-person language may detract from the arguments and/or viewpoints present in the writing. Therefore, avoid using first-person frequently, you may use passive voice instead

Eg. ‘We apply k-means cluster analysis to classify the NDSs based on similarities’

Suggest rewording as follows.

NDSs based on similarities were classified using k-means cluster analysis.

Response 2:

Sentences that were in active voice / first-person pronoun in the original draft are transformed into passive voice, at all instances in the manuscript.

Response to the Comments of Reviewer 1

Comment 1:

The authors have attempted to investigate the important issue of how well the national dementia strategies (NDSs) of 15 European countries align with the Glasgow Declaration and the WHO’s Global Action Plan guidance. However, the current version of the article does not meet academic standards, as information relevant to the introduction, methods, and results is interwoven across all sections. For example, the introduction includes methodological details, and vice versa. I strongly recommend that the authors carefully revise the text to ensure that each section is structured appropriately.

Response 1:

The manuscript is restructured to ensure that there are no overlaps of contents across sections, and each section contains the text relevant to the said section. The redundancies that have arisen because of the overlap in the original draft are also removed.

Additionally, Objectives of the study are now, explicitly stated in the subsection of the Methods section

Comment 2:

Although the primary objective of the study is to examine how well the selected NDSs comply with the Glasgow Declaration and the WHO’s Global Action Plan, the article does not directly compare the NDSs with these benchmark criteria. Furthermore, the discussion is not structured around the key action areas outlined in the WHO framework or the Glasgow Declaration.

Response 2:

Added two sub-sections, in the Discussion section, to highlight the extent of adherence of NDSs to the fundamental rights of Glasgow Declaration and the key action areas of the WHO’s global action plan.

Comment 3:

It is also unclear why more than half of the discussion focuses on the costs and cost-effectiveness of dementia interventions, given that these were not the primary focus of the study.

Response 3:

The discussion on cost-effectiveness of the provisions in the national dementia strategies is abridged and put in perspective of the NDSs’ compliance with Glasgow Declaration and WHO’s Global Action Plan.

Comment 4:

Additionally, a clear summary of findings is missing, and the results are not interpreted in relation to the WHO key action areas or the Glasgow Declaration.

Response 4:

Conclusion section is revised to present the summary of findings.

Additionally

A section on Limitations of this Study is added, which was discussed as a part of the Discussion section in the original draft.

Abstract is edited to reflect the changes made in the manuscript.

Journal Requirements

Comment 1:

Please ensure that your manuscript meets PLOS ONE's style requirements, including those for file naming. The PLOS ONE style templates can be found at https://journals.plos.org/plosone/s/file?id=wjVg/PLOSOne_formatting_sample_main_body.pdf and https://journals.plos.org/plosone/s/file?id=ba62/PLOSOne_formatting_sample_title_authors_affiliations.pdf

Response 1:

Title page is now, formatted as per the guidelines shared.

Headings are now, formatted as per the guidelines shared.

Comment 2:

We note that the grant information you provided in the ‘Funding Information’ and ‘Financial Disclosure’ sections do not match.

Response 2:

There are no grant numbers associated with the funding. However, I have revised the funding statement as under:

Revised Funding Statement

Smruti Bulsari, University of Essex, is supported by the National Institute for Health and Care Research – Applied Research Collaboration East of England (NIHR ARC EoE) and the Alzheimer’s Society, funded through a Post-Doctoral Fellowship. The NIHR ARC EoE is hosted by Cambridgeshire and Peterborough NHS Foundation Trust. The views expressed are those of the authors and not necessarily those of the NIHR or the Department of Health and Social Care.

Comment 3:

Thank you for stating the following in the Competing Interests section:

I have read the journal's policy and the authors of this manuscript have the following competing interests: One of the co-authors (RK) is currently serving on the editorial board (academic editor) of PLOS One.

Please confirm that this does not alter your adherence to all PLOS ONE policies on sharing data and materials, by including the following statement: "This does not alter our adherence to PLOS ONE policies on sharing data and materials.” (as detailed online in our guide for authors http://journals.plos.org/plosone/s/competing-interests).

Response 3:

Updated Competing Interest Statement:

I have read the journal's policy, and the authors of this manuscript have the following competing interests: One of the co-authors (RK) is currently serving on the editorial board (academic editor) of PLOS One. This does not alter our adherence to PLOS ONE policies on sharing data and materials.

Comment 4:

Thank you for stating the following financial disclosure:

This study is funded by National Institute of Health and Care Research (NIHR) Applied Research Collaboration (ARC) East of England and Alzheimer's Society

Response 4:

Updated Financial Disclosure

This study is funded by National Institute of Health and Care Research (NIHR) Applied Research Collaboration (ARC) East of England and Alzheimer's Society. The funders had no role in study design, data collection and analysis, decision to publish, or preparation of the manuscript.

Comment 5:

Thank you for stating the following in the Acknowledgments Section of your manuscript:

Smruti Bulsari, University of Essex, is supported by the National Institute for Health and Care Research – Applied Research Collaboration East of England (NIHR ARC EoE) and the Alzheimer’s Society, funded through a Post-Doctoral Fellowship. The NIHR ARC EoE is hosted by Cambridgeshire and Peterborough NHS Foundation Trust. The views expressed are those of the authors and not necessarily those of the NIHR or the Department of Health and Social Care.

Please remove any funding-related text from the manuscript and let us know how you would like to update your Funding Statement.

Response 5:

Acknowledgement section is now removed, in the revised manuscript, and the Funding Statement is revised as in response 2 (to comment 2) under the heading Journal requirements (above).

Comment 6:

Please include captions for your Supporting Information files at the end of your manuscript, and update any in-text citations to match accordingly. Please see our Supporting Information guidelines for more information: http://journals.plos.org/plosone/s/supporting-information.

Response 6:

Created four separate files for supplementary information and details given along with caption and one-line title, at the end of the manuscript. In-text references are edited and collated accordingly.

Finally, with reference to the recommendation “While revising your submission, please upload your figure files to the Preflight Analysis and Conversion Engine (PACE) digital diagnostic tool, https://pacev2.apexcovantage.com/. PACE helps ensure that figures meet PLOS requirements.”

Response: Figure files uploaded to PACE V2.

Also, figures have been renamed in the manuscript, to match the renamed figures as per PLOS guidelines by PACE V2. All corresponding references to the figure in the manuscript are also renamed.

---

## [Decision Letter · Decision Letter 1]

30 Jul 2025

PONE-D-25-06352R1A Comparative Analysis of Dementia Strategies of Fifteen European Countries in the Context of Glasgow Declaration and WHO’s Global Action PlanPLOS ONE

Dear Dr.Smruti,

Thank you for submitting your manuscript to PLOS ONE. After careful consideration, we feel that it has merit but does not fully meet PLOS ONE’s publication criteria as it currently stands. Therefore, we invite you to submit a revised version of the manuscript that addresses the points raised during the review process.

We look forward to receiving your revised manuscript.

Kind regards,

Surangi Jayakody, MBBS, MSc, MD

Academic Editor

PLOS ONE

Journal Requirements:

Reviewers' comments:

Reviewer's Responses to Questions

**Comments to the Author**

1. If the authors have adequately addressed your comments raised in a previous round of review and you feel that this manuscript is now acceptable for publication, you may indicate that here to bypass the “Comments to the Author” section, enter your conflict of interest statement in the “Confidential to Editor” section, and submit your "Accept" recommendation.

Reviewer #1: (No Response)

Reviewer #2: All comments have been addressed

2. Is the manuscript technically sound, and do the data support the conclusions?

Reviewer #1: Partly

Reviewer #2: Yes

3. Has the statistical analysis been performed appropriately and rigorously? 

Reviewer #1: N/A

Reviewer #2: N/A

4. Have the authors made all data underlying the findings in their manuscript fully available?

Reviewer #1: Yes

Reviewer #2: Yes

5. Is the manuscript presented in an intelligible fashion and written in standard English?

Reviewer #1: Yes

Reviewer #2: Yes

6. Review Comments to the Author

Reviewer #1: Thank you for the revised submission. The authors have made substantial improvements to the manuscript, and I commend them for their careful attention to the reviewer feedback.

The structural issues raised in the initial review have been fully addressed — the Introduction, Methods, Results, and Discussion are now clearly delineated and logically presented. The summary and interpretation of findings have also been strengthened, with clearer thematic insights and better alignment with the WHO Global Action Plan and Glasgow Declaration. In addition, typographical and grammatical errors have been corrected, improving clarity and readability throughout.

A few points were only partially addressed. While the discussion now engages more meaningfully with the WHO and Glasgow benchmarks, an explicit comparison table or matrix would have added further clarity. Similarly, although the discussion on cost-effectiveness is now better contextualised, it remains relatively long compared to the main findings and could be further shortened.

Reviewer #2: The authors investigated on how well the National Dementia Strategies (NDSs) of 15 European countries align with the Glasgow Declaration and WHO’s Global Action Plan(GAP).

However, there are some areas to be revisited and could have written in a more concise manner.

Authors mentioned the objectives in the Methods section. The objectives should be at the end of the justification and before the “Methods” section

Methods –

It is better if the methods are described in a concise way which all readers can grab how the study is conducted clearly. Also, it is great if the authors mention, study period which helps the reader to understand by which year these strategic plans are considered for the study.

Table 1 (Page 21) depicts the cluster membership of countries based on NDSs contents. This is an output from the analysis by the authors for sure; According to this table, there are 17 countries categorized into clusters whereas for the study, there are only 15 countries included. How can the “Denmark” and “Greece” come into play additionally while the NDSs of those two countries are not included in the study?

In line 255 also, the authors mentioned “Eight out of 17 countries have adopted the Beveridge model of healthcare………….” I wonder why the authors referred to 17 countries instead of 15??

The authors presented the comparison of NDSs with Glasgow Declaration and KAA of WHO’s Global Action Plan in the “discussion” section. I wonder if these are the main parts to be included in the “results” section. Also, the other findings of the results should have been presented in a concise manner.

In the methods, the authors explained fiscal contextualization by examining different models. However, it is difficult to find any interrelation of financial implications of fundamental rights of Glasgow Declaration& KAA of WHO GAP with present study findings in the discussion section. Is this an additional area that the authors are looking into other than the objectives?

7. PLOS authors have the option to publish the peer review history of their article (what does this mean? ). If published, this will include your full peer review and any attached files.

**Do you want your identity to be public for this peer review?** For information about this choice, including consent withdrawal, please see our Privacy Policy .

Reviewer #1: No

Reviewer #2: No

---

## [Author Response · Author response to Decision Letter 2]

10 Sep 2025

I would like to thank the reviewers for their time for for meticulously going through our revision, and your valuable suggestions to make our work more accessible and comprehensive for the prospective readers.

Response to the comments of Reviewer 1

Comment 1: While the discussion now engages more meaningfully with the WHO and Glasgow benchmarks, an explicit comparison table or matrix would have added further clarity. Similarly, although the discussion on cost-effectiveness is now better contextualised, it remains relatively long compared to the main findings and could be further shortened.

Regarding comparison matrix: A matrix is developed and placed in the results section under the heading “Comparison of NDSs clusters with the Fundamental Rights of the Glasgow Declaration and the KAAs of the WHO’s Global Action Plan”, along with a brief explanation.

Because this section now makes a comparison of the NDSs with the fundamental rights of the Glasgow Declaration and the KAAs of the Global Action Plan, the titles of sub-section in the discussion section are changed as:

Revised Title: “NDSs Clusters in the context of the Fundamental Rights of the Glasgow Declaration”

Original Title: “Comparison of NDSs Clusters with the Fundamental Rights of the Glasgow Declaration”, and

Revised Title: “NDSs Clusters in the context of the Key Action Areas of WHO’s Global Action Plan”

Original Title: “Comparison of NDSs Clusters with the Key Action Areas of WHO’s Global Action Plan”

Text has been edited in these sub-sections of the discussion section to remove redundancies with the newly added section on Comparison of NDSs clusters with the “Fundamental Rights of the Glasgow Declaration and the KAAs of the WHO’s Global Action Plan”.

Regarding the discussion on cost-effectiveness: It is now shortened, while retaining its essence and attempting to make it even more contextual.

Response to the comments of Reviewer 2

Comment 1: Authors mentioned the objectives in the Methods section. The objectives should be at the end of the justification and before the “Methods” section

Response 1: The objectives are now, moved to the end of the justification.

Comment 2: It is better if the methods are described in a concise way which all readers can grab how the study is conducted clearly. Also, it is great if the authors mention, study period which helps the reader to understand by which year these strategic plans are considered for the study.

Response 2: Methods section is revised to be presented in a concise form.

Regarding study period, this text is added at the end of the second paragraph of “Data for this Study….” Section. The text explains why it is difficult to explicitly specify the time-period of this study: “…Therefore, this makes it difficult to explicitly specify the time-period of this study. Though, the oldest dementia strategy is of Cyprus, which is published in 2012 and the most recent is Germany’s, published in 2023; the dementia strategies used in this study were downloaded in November 2023; the year for the NDS of each of these countries are given in are Table s1”.

Comment 3: Table 1 (Page 21) depicts the cluster membership of countries based on NDSs contents. This is an output from the analysis by the authors for sure; According to this table, there are 17 countries categorized into clusters whereas for the study, there are only 15 countries included. How can the “Denmark” and “Greece” come into play additionally while the NDSs of those two countries are not included in the study?

Response 3: We are extremely sorry for this oversight. We rechecked the data and there are 17 countries. We have included the NDSs of Denmark and Greece in our analysis. Therefore, we have replaced 17 in place of 15 European countries, everywhere in the manuscript, abstract and the title. We rechecked and confirmed that in all our outputs, there are 17 European countries, which includes Greece and Denmark. Therefore, in the methods section too, we have added these two countries, which were missing because of our oversight.

Other relevant details are also edited to ensure clarity and eliminate contradictions, if any.

Comment 4: In line 255 also, the authors mentioned “Eight out of 17 countries have adopted the Beveridge model of healthcare………….” I wonder why the authors referred to 17 countries instead of 15??

Response 4: The response to comment 3 explains this.

Comment 5: The authors presented the comparison of NDSs with Glasgow Declaration and KAA of WHO’s Global Action Plan in the “discussion” section. I wonder if these are the main parts to be included in the “results” section. Also, the other findings of the results should have been presented in a concise manner.

Response 5: A section on comparison is now presented in a concise manner under the results section, along with a comparison matrix. The titles of these sub-sections in the discussion section are changed as:

Revised Title: “NDSs Clusters in the context of the Fundamental Rights of the Glasgow Declaration”

Original Title: “Comparison of NDSs Clusters with the Fundamental Rights of the Glasgow Declaration”, and

Revised Title: “NDSs Clusters in the context of the Key Action Areas of WHO’s Global Action Plan”

Original Title: “Comparison of NDSs Clusters with the Key Action Areas of WHO’s Global Action Plan”

The focus of these sub-sections is to highlight the intricacies of the fundamental rights / KAAs that are adhered, partially adhered or not adhered to and in what way there are adhered to, for each of the NDSs clusters, and which ones require attention in the NDSs. Also, the sub-section on “NDSs Clusters in the context of the Key Action Areas of WHO’s Global Action Plan” is substantially revised.

Comment 6: In the methods, the authors explained fiscal contextualization by examining different models. However, it is difficult to find any interrelation of financial implications of fundamental rights of Glasgow Declaration& KAA of WHO GAP with present study findings in the discussion section. Is this an additional area that the authors are looking into other than the objectives?

Response 6: In the context of the above two sections (renamed as explained in comment 5), which highlight the areas that need to be addressed by the NDSs, the interrelation of financial implications of fundamental rights of Glasgow Declaration& KAA of WHO GAP is explained in the discussion section. Also, the discussion on financial implications is shortened. A statement in conclusion section is also added to highlight its relevance.

---

## [Editor Report · Decision Letter 2]

6 Oct 2025

A Comparative Analysis of Dementia Strategies of Seventeen European Countries in the Context of Glasgow Declaration and WHO’s Global Action Plan

PONE-D-25-06352R2

Dear Dr. Bulsari,

We’re pleased to inform you that your manuscript has been judged scientifically suitable for publication and will be formally accepted for publication once it meets all outstanding technical requirements.

Kind regards,

Surangi Jayakody, MBBS, MSc, MD

Academic Editor

PLOS ONE
---

## [Editor Report · Acceptance letter]

PONE-D-25-06352R2

PLOS ONE

Dear Dr. Bulsari,

I'm pleased to inform you that your manuscript has been deemed suitable for publication in PLOS ONE. Congratulations! Your manuscript is now being handed over to our production team.

Kind regards,

on behalf of

Dr Surangi Jayakody

Academic Editor

PLOS ONE